# Health related quality of life of people receiving highly active antiretroviral therapy in Southwest Ethiopia

**Addisu Desta[1]⊕, Tessema Tsehay Biru[2]⊕, Adane Teshome Kefale ⓘ[3,4]⊕ ***

**1** Department of Pharmacy, Wachemo University, Hossaena, Ethiopia, **2** Department of Pharmacy, Wollo University, Dessie, Ethiopia, **3** Department of Pharmacy, Debre Berhan University, Debre Berhan, Ethiopia, **4** School of Pharmacy and Pharmacology, University of Tasmania, Hobart, Australia

⊕ These authors contributed equally to this work.
* adane@mtu.edu.et

## Abstract

### Background

Highly Active Antiretroviral Therapy (HAART) is a standard of HIV management to suppress viral load and delay progression to AIDS. However, questions have been raised about the use of antiretroviral therapy and how it affects quality of life (QoL) of people living with HIV/AIDS (PLWHA). The study hence aimed to assess the QoL of PLWHA who were taking HAART at Mizan–Tepi University Teaching Hospital (MTUTH) and identify factors associated with QoL.

### Methods

A cross sectional study was conducted among PLWHA receiving HAART at MTUTH from March 04-April 1, 2018. Patients were recruited consecutively and interviewed with structured questionnaire. A data abstraction tool was used to extract data from patient medical records. Quality of life was assessed using the World Health Organization Quality of Life HIV- BREF (WHOQOL-HIV-BREF) standard tool. Data was entered to Epi-Info version 3.5.3 and analyzed using SPSS version 22 for windows. A multivariable logistic regression analysis was fitted to identify factors associated with QoL. A statistical significance was established at a p value <0.05.

### Results

A total of 240 participants with the mean age of 35.11 (SD = 9.08) years were included in the study. This study found that 57.1% of the patients had high global score of QoL. Patients with normal current health (AOR = 3.38, 95% CI = 1.56–7.31)) and having family support (AOR = 3.12, 95% CI = 1.51–6.46) were positively associated with high global score of QoL, while patients with low HAART adherence (AOR = 0.40, 95%, CI = 0.19–0.86) were negatively associated with high global score of QoL.

**Data Availability Statement:** All relevant data are within the manuscript and its Supporting Information files.

**Funding:** The authors received no specific funding for this work.

**Competing interests:** The authors have declared that no competing interests exist.

## Conclusion

The study revealed that more than half of the participants had high global score of QoL. Normal current health and family support were associated with better global score of QoL, while low HAART adherence was found to be associated with the lower global score of QoL.

## Introduction

Human Immunodeficiency Virus (HIV) primary affects the immune system that predisposes the victim to multiple opportunistic diseases leading to Acquired Immuno-deficiency Syndrome (AIDS) [1]. According to UNAIDS, there were approximately 36.7 million people worldwide living with HIV/AIDS at the end of 2016 [2]. Most HIV infected people, approximately 95% of the total, live in developing countries. Sub-Saharan Africa has been hit especially hard, with almost 70% of all HIV-infected patients living there [3]. Adult HIV prevalence in Ethiopia was estimated to be 1.1% in 2016 [4].

Introduction of Highly Active Anti-Retroviral Therapy (HAART) was a major turning point in HIV care. It uses a combination of antiretroviral medication recommended to aggressively suppress viral replication and halt progress of HIV to AIDS, hence a subsequent improvement in survival and quality of life (QoL) [5–8]. Significant efforts have been exerted to scale-up HAART uptake in developing countries; particularly in sub–Saharan Africa, where the epidemic has had its most devastating impact. However, there are concerns about impact of HAART on the QoL of people living with HIV/AIDS (PLWHA), where it is influenced by multiple factors [9–11]. According to the World Health Organization (WHO), QoL is defined as individuals' perceptions of their position in life in the context of the culture and value systems in which they live and in relation to their goals, standards, expectations and concerns they have [12]. This definition implies that QoL is largely dependent on cultural, social and environmental contexts as well as individual perception [12, 13]. QoL has been used as a criterion in assessing HIV/AIDS prevention programs, clinical treatment, and harm reduction strategies [14]. Multilevel interventions and long-term care, rehabilitation, behavioral therapy and social supports for patients receiving treatments, and application of e-health approaches can improve QoL of HIV patients [15].

Although HAART is usually considered as a standard of care and savior for PLWHA [8, 16], QoL of people taking Anti-retroviral Therapy (ART) remains a concern. Healthcare is dynamic; in continuous change, aimed to increase the length and quality of survival. Consequently, to increase the length and quality of survival of PLWHA who are on ART, it is essential that an evaluation of their QoL becomes more important than quantity of life. ART may indeed prolong life but may only do so at considerable cost to the QoL of PLWHA [17]. There is a geographical discrepancy in HIV/AIDS economic evaluation research [18]. The high cost of ART might reduce the adherence to treatment and worsen QoL [19]. An improved QoL and resultant ability of the patient to resume normal life, including supporting the families and working productively will encourage long-term maintenance of treatment. However, if QOL is poor, it impacts negatively on lifelong adherence to medication [20].

Through reflection on the sense of well-being and satisfaction experienced by people under their current life circumstances, the assessment of QoL aims to provide a comprehensive evaluation of the individual's well-being, which includes an assessment of their role functioning, community integration and personal adjustment [21]. Previously, studies were conducted in different parts of Ethiopia [9, 22–26], but data are scanty on QoL of PLWHA in the current

setting. Since QoL is affected by many factors that varies from setting to setting, it is important to evaluate the QoL at this area. Thus, the study aimed to assess the QoL and its determinants among PLWHA who were receiving HAART at Mizan–Tepi University teaching hospital (MTUTH).

## Methods

### Study setting and design

A facility based cross-sectional study was conducted at the ART clinic of MTUTH, located in Mizan-Aman town, Southwest of Ethiopia. It provides services to the residents of Benchi Maji zone and Gambella Regional state. At the time of data collection, 1639 patients were actively receiving HAART at the hospital. Data was conducted from March 04-April 1, 2018.

### Sampling procedure

The sample size was calculated using the single population proportion formula. Considering a z value of 1.96 for 95% confidence interval, 50% prevalence of poor QoL and 5% of margin of error, gives initial sample size of 384. The initial sample was adjusted using a correction formula for the study population (population<10,000) giving the final sample size of 311.

The study included PLWHA aged $\geq$ 18 years old who were on HAART for at least 6 months. Patients with incomplete medical records (lack of CD4 count or absence of WHO clinical stage), and those not willing to participate were excluded from study.

All patients who visited the ART Pharmacy during data collection period, and who fulfilled inclusion criteria were recruited for the study. Due to low patient flow, unwillingness of participants, and incomplete medical records, only 240 patients were included in the final analysis.

### Data collection instrument and process

The WHOQOL-HIV BREF was used to assess the QoL of PLWHA receiving ART. Briefly, the tool consisted of 31 items/facets. Of 31 items, 29 items were used to measure individual QoL across six domains while two items were used to measure patients' perception of their general QoL and health status. The six domains are physical, psychological, level of independence, social relationships, environmental, and spirituality, religion, personal beliefs (SRPB). Each item is rated in a five-point Likert scale where 1 indicates low, negative perception and 5 indicate high, positive perception. However, the score for pain and discomfort, dependence of medication, death and dying, and negative feelings and other negatively phrased items were reversed using the formula 6-X (where X was the facet score) [12, 27, 28].

The facet score was calculated by dividing a sum of all items of the facet by four. The facet scores within each domain were used to calculate the domain score. Hence, the domain scores were computed by multiplying the mean of all items within the domain by four. A global score of QoL was similarly calculated using the mean scores of all the six domains. All domain scores ranged from 4 to 20 [12]. Higher scores in each domain indicated higher QoL for that domain [28]. Patients' QoL was categorized as poor if their score is below the sample mean, and good when they scored equal or greater than the mean. The overall QoL of each patient was classified as high or low taking the sample mean global score as a reference. The tool was translated into the local language (Amharic) and was previously validated and widely used in other settings in Ethiopia [22, 24–26, 29].

Adherence to HAART was assessed by a standard Morisky Medication Adherence Predictor Scale (MMAPS-8) designed for adherence measurement in chronic diseases [30, 31]. The tool was widely used to assess medication adherence in PLWHA in Ethiopia [32, 33]. The scale

has eight items which are used to assess patient's HAART adherence over the past two weeks. Each question has a response of Yes (1) or No (0) with score for item number 5 being reversed. Accordingly, patients with a score of 0, 1–2 and ≥3 to the MMAPS-8 are categorized to have high, moderate, and low levels of HAART adherence, respectively. A data abstraction format was also used to collect patients' clinical information from their medical records.

Data was collected by trained pharmacists. Patients were approached at the end of medication fill at the ART pharmacy. An interviewer administered questionnaire was used instead of self-administered due to inclusion of illiterate participants in the study. First the interview was carried out for each participants and clinical information was extracted from their respective medical records.

The questionnaire and data abstraction format were checked thoroughly for comprehensiveness before commencement of the actual data collection through pre-test. The data collector made frequent checks on the data collection process to ensure data quality. The collected data was checked for its completeness, accuracy, clarity, and consistency after conducting data collection. Pre-test was done on 10 patients.

### Data entry and analysis

The collected data was entered to Epi Info version 3.5.3 and analyzed using Statistical Package for Social Sciences (SPSS) version 22.0 for Windows. Descriptive statistics (frequency, mean, SD) were used for frequency distributions of responses.

A binary logistic regression analysis was undertaken to identify factors associated with high global score of QoL. Accordingly, crude odd ratio (COR) and adjusted odd ratio (AOR) were used to measure level of association during the bivariate and multivariable analysis, respectively. A p value < 0.05 was used to declare a statistically significant association at 95% of confidence interval (CI).

### Ethical consideration

The study was approved by institutional review board of College of Health Sciences, Mizan-Tepi University. A formal letter of cooperation was submitted to ART clinic of the hospital and permission was secured before data collection. All patients were informed about the purpose of the study, the importance of their participation and verbal consent was taken, and interview were proceeded after the consent. The consent was recorded in each questionnaire. Participation in the study was fully voluntarily and informed about their right to leave the study at any time. No personal identifiers were included in the data and confidentiality of the collected data was always maintained.

## Results

### Sociodemographic characteristics of patients

We have approached 311 patients, but data of 240 was included in the analysis giving a response rate of 77.2% (excluded due to unwillingness to participate, incomplete medical records, and lost medical records). A total of 240 patients with mean age of 35.11 ±9.08 years; ranged from 18 to 62 years were included in the final analysis. Females comprised 63.8% of participants. Majority of the patients (132, 55%) completed primary education and 130(54.2%) patients were married (Table 1).

**Table 1. Socio-demographic characteristics of PLWHA on HAART at MTUTH, March 2018.**

| Variable | Category | Freq. | Percentage |
|---|---|---|---|
| Sex | Male | 87 | 36.25 |
| | Female | 153 | 63.75 |
| Age (Years) | ≤ 30 | 87 | 36.25 |
| | 31–40 | 99 | 41.25 |
| | >40 | 54 | 22.50 |
| Education | Cannot Read and Write | 39 | 16.25 |
| | Primary (1–8 Grade) | 132 | 55.00 |
| | Secondary (9–12 Grade) | 50 | 20.83 |
| | Higher Education | 19 | 7.90 |
| Marital Status | Single | 16 | 6.70 |
| | Married | 130 | 54.20 |
| | Divorce | 62 | 25.83 |
| | Widowed | 32 | 3.30 |
| Occupation | Farmer | 15 | 6.25 |
| | Gov't Employee | 42 | 17.50 |
| | Trade/Private | 56 | 23.33 |
| | Daily Labourer | 38 | 15.83 |
| | House Wife | 50 | 20.83 |
| | Unemployed | 23 | 9.58 |
| | Other* | 16 | 6.68 |
| Residency | Rural | 34 | 14.17 |
| | Urban | 206 | 85.83 |
| Having Children | No | 49 | 20.42 |
| | Yes | 191 | 79.58 |

*Carpenter (2), Driver (4), Mechanic (1), Student (6), Maid (1), Retired (2)

## Self-perceived health status and other health related information

Patients were asked to rate their current health status; accordingly, 26.3%, 20% and 2.1% rated their current health as very good, neither good nor bad, and very poor, respectively. Approximately two-third of (64.6%) could not ascertain the source of HIV infection, while 71 (29.6%) patients admitted to a sexual relationship as the source of infection. Most patients (86.25%) had disclosed their HIV status to their relatives. Regarding support, more than half (63.33%) of the participants reported that they had family support (Table 2).

Seventeen patients were found to be the current substance users, including Khat and alcohol. Adherence to HAART was assessed using MMAPS-8. Accordingly, out of 240 patients, 187 (77.90%), 20 (8.30%) and 33 (13.80%) patients had high, medium, and low HAART adherence, respectively (Table 3).

## Clinical information

At baseline, the mean body weight of participants was 51.83 ± 9.55 Kg, while at the time of data collection, 55.59±10.00 Kg. Most of the participants (57.81%) had a normal BMI, with mean of 20.48±3.58 Kg/m². Only 33.47% of patients had baseline $CD_4$ count $\geq$350 cells/mm³ (mean: 333.54±294.77), and 97.87% of them had the most recent $CD_4$ count $\geq$100 cells/mm³(mean: 594.63±347.28)). More than half of the patients were in advanced stages of HIV (stage III and IV) during enrollment to ART care, while most of them (87.08%) had a current

**Table 2. Self-perceived health status and other health related information of PLWHA on HAART at MTUTH, March 2018.**

| Variable | Category | Frequency | Percentage |
|---|---|---|---|
| Perceived current health status | Very poor | 5 | 2.08 |
| | Poor | 14 | 5.83 |
| | Neither Poor nor Good | 48 | 20.00 |
| | Good | 110 | 45.83 |
| | Very Good | 63 | 26.25 |
| Current ill Health | No | 171 | 71.25 |
| | Yes | 69 | 28.75 |
| Believe you were infected with HIV | Unprotected Intercourse | 71 | 29.58 |
| | Blood products | 14 | 5.83 |
| | Unknown | 155 | 64.58 |
| Duration since tested +Ve for HIV (Years) | ≤ 2 | 150 | 62.50 |
| | >2 | 90 | 37.50 |
| Disclosure HIV status to relatives | No | 33 | 13.75 |
| | Yes | 207 | 86.25 |
| Family members with HIV | No | 130 | 54.17 |
| | Yes | 110 | 45.83 |
| Number of family members affected with HIV (n = 110) | 1 | 89 | 80.91 |
| | 2 | 15 | 13.64 |
| | ≥3 | 6 | 5.45 |
| Living environment | Very bad | 7 | 2.92 |
| | Bad | 18 | 7.50 |
| | Neither bad nor Good | 64 | 26.67 |
| | Good | 124 | 51.67 |
| | Very Good | 27 | 11.25 |
| Family support | No | 88 | 36.67 |
| | Yes | 152 | 63.33 |
| Social relationship | Very bad | 1 | 0.42 |
| | Bad | 8 | 3.33 |
| | Neither bad nor Good | 9 | 3.75 |
| | Good | 107 | 44.58 |
| | Very Good | 115 | 47.92 |
| Recent experience with HIV stigma and discrimination | No | 222 | 92.50 |
| | Yes | 18 | 7.50 |

HIV: Human Immunodeficiency Virus Duration since tested positive for HIV (Months): Mean (SD) = 79.55 (44.55) Range (Min-Max) = 198(6–204)

WHO clinical stage I disease. Among 175 patients for whom the most recent viral load was recorded, 88.57% had a viral load measurement of ≤1000 copies/ml (mean: 426.86±1334.52). More than half (144, 60%) of the patients-initiated ART with the first line regimen (TDF+3TC +EFV), with 184 (76.67%) remained on the same ART regimen during their treatment. Among those patients who made a regimen change (N = 56), development of toxicity (48.21%) was accounted as a major reason (Table 4).

## HRQOL: Mean scores, level of mean scores and global sores of QoL domains

The HRQOL was assessed using WHOQOL-HIV BREF. The internal consistency of the tool was assessed using Cronbach's alpha coefficient. The analysis showed that alpha value of 0.897,

**Table 3. Health related information and drug taking behaviour of PLWHA on HAART MTUTH, March 2018.**

| Variable | Category | Frequency | % |
|---|---|---|---|
| Perceived baseline quality of life | Very Bad | 34 | 14.17 |
| | Bad | 46 | 19.17 |
| | Neither Bad nor Good | 31 | 12.92 |
| | Good | 96 | 40.00 |
| | Very Good | 33 | 13.75 |
| Ever used any substance | No | 166 | 69.17 |
| | Yes | 74 | 30.83 |
| Currently using any substance (n = 74) | No | 57 | 77.02 |
| | Yes | 17 | 22.97 |
| Type of current substance use (n = 17) | Alcohol | 6 | 35.29 |
| | Khat | 11 | 64.71 |
| Frequency of ART Administration per day | Once | 172 | 71.67 |
| | Twice | 68 | 28.33 |
| Regular follow up for HIV | No | 1 | 0.42 |
| | Yes | 239 | 99.58 |
| Relation with health care provider | Very Bad | 1 | 0.42 |
| | Bad | 1 | 0.42 |
| | Neither Bad nor Good | 3 | 1.25 |
| | Good | 60 | 25.00 |
| | Very Good | 175 | 72.90 |
| Level of HAART Adherence | High | 187 | 77.90 |
| | Medium | 20 | 8.30 |
| | Low | 33 | 13.80 |

ART: Antiretroviral Therapy; HAART: Highly Active Antiretroviral Therapy; HIV: Human Immunodeficiency Virus

which revealed the internal reliability of the questionnaire, as it is higher than the recommended cut-off point (alpha > 0.7) [34].

The mean scores of each facet of QoL domains and the two general facets were determined. The higher the score of the facets, the better global score of QoL. The scores for seven facets were reversed so that the lower the score indicates the higher global score of QoL. The mean scores of QoL were high in the SRPB (17.13), physical (16.34) and level of independence (16.21) domains, while it was medium in the remaining domains (Table 5).

The mean scores of QoL were categorized into low and high scores for each domain. Among the six domains, the largest proportions of participants scored a high QoL in SRPB domain (64.6%) followed by psychological domain (62.5%). In contrary, a higher percentage of participants (57.5%) scored low QoL in social relationship domain (Fig 1).

About 57.1% of the participants had a high global score of QoL, while the remain 42.9% had low global score of QoL.

## Factors associated with poor quality of life

A bivariate binary logistic regression analysis was run to identify any association between different sociodemographic and clinical variables with global score of QoL of the participants. Among sociodemographic variables, educational status, occupation, marital status, and family support were associated with high global score of QoL. In addition, HAART adherence, current ill health, and BMI were clinical variables associated with global score of QoL.

**Table 4. Clinical information of PLWHA on HAART at MTUTH, March 2018.**

| Variable | Category | Frequency | Percent |
|---|---|---|---|
| Recent BMI (n = 237) | Under weight (BMI<18.5) | 74 | 31.22 |
| | Normal BMI (18.5–24.99) | 137 | 57.81 |
| | Overweight (25–24.99) | 26 | 10.97 |
| Baseline CD$_4$ Count (n = 239) | <350 | 159 | 66.53 |
| | ≥350 | 80 | 33.47 |
| Recent CD$_4$ Count (n = 235) | <100 | 5 | 2.13 |
| | ≥ 100 | 230 | 97.87 |
| Baseline VL (n = 20) | ≤ 1000 | 11 | 55.00 |
| | > 1000 | 9 | 45.00 |
| Recent VL (n = 175) | ≤ 1000 | 155 | 88.57 |
| | > 1000 | 20 | 11.43 |
| Baseline WHO Clinical Stage | Stage I | 66 | 27.50 |
| | Stage II | 48 | 20.00 |
| | Stage III | 114 | 47.50 |
| | Stage IV | 12 | 5.00 |
| Recent WHO Clinical Stage | Stage I | 209 | 87.08 |
| | Stage II | 24 | 10.00 |
| | Stage III | 7 | 2.92 |
| Comorbidity (at least one) | No | 229 | 95.42 |
| | Yes | 11 | 4.58 |
| Type of comorbidity (Total comorbidity = 12) | Asthma | 4 | 33.33 |
| | Chronic Kidney Disease | 3 | 25.00 |
| | Hypertension | 2 | 16.67 |
| | Heart Failure | 2 | 16.67 |
| | Major Depressive Disorder | 1 | 8.33 |
| OIs (at least one) | No | 172 | 71.67 |
| | Yes | 68 | 28.33 |
| Type of OIs (Total OIs = 68) | Tuberculosis | 67 | 98.53 |
| | Toxoplasmosis | 1 | 1.47 |
| Cotrimoxazole preventive therapy | No | 42 | 17.50 |
| | Yes | 198 | 82.50 |
| Isoniazid preventive therapy | No | 50 | 20.83 |
| | Yes | 190 | 79.17 |
| Type of Initial HAART regimen | D4T-3TC-NVP | 25 | 10.42 |
| | D4T-3TC-EFV | 8 | 3.33 |
| | AZT-3TC-NVP | 31 | 12.92 |
| | AZT-3TC-EFV | 21 | 8.75 |
| | TDF-3TC-EFV | 144 | 60.00 |
| | TDF-3TC-NVP | 7 | 2.92 |
| | other* | 4 | 1.67 |
| Total duration on HAART (Years)** | ≤ 1 | 51 | 21.25 |
| | >1–5 | 75 | 31.25 |
| | >5–10 | 96 | 40.00 |
| | >10 | 18 | 7.50 |
| Initial Regimen change | No | 184 | 76.67 |
| | Yes | 56 | 23.33 |
| Reasons for regimen change (n = 56) | Toxicity | 27 | 48.21 |
| | Treatment failure | 13 | 23.21 |
| | Not documented | 12 | 21.43 |
| | New Tuberculosis | 3 | 5.36 |
| | Drug non-availability | 1 | 1.79 |

(*Continued*)

**Table 4.** (*Continued*)

| Variable | Category | Frequency | Percent |
|---|---|---|---|
| ADRs | No | 181 | 75.42 |
| | Yes | 59 | 24.58 |

ADRs: Adverse Drug Reactions; BMI: Body Mass Index (Kg/m$^2$); OIs: Opportunistic Infections; VL: Viral Load (Copies/ml).

* 2F (AZT-3TC-ATV/r), 4d (AZT-3TC-EFV for child), 4a (d4t-3TC-NVP for child), 4C (AZT-3TC-NVP for child)

** Total duration on HAART (Years): Mean (SD) = 5.55(3.59)

**Table 5. Mean scores for each of the six QoL domain facets for PLWHA on HAART at MTUTH, March 2018.**

| Domains | Facets | Mean (SD) | Domain mean score (out of 20) |
|---|---|---|---|
| General | Rating Quality of Life | 3.84(0.88) | |
| | Satisfaction with health | 3.75(0.95) | |
| Physical | Extent to which a patient feels that physical pain prevents from doing what is needed to be done? (REVERSED) | 4.23(1.25) | 16.34(3.36) |
| | Extent a patient is bothered by any physical problems related to HIV infection (REVERSED) | 4.29(1.22) | |
| | Having enough energy for everyday life | 3.74(1.31) | |
| | Satisfaction with sleep | 4.08(1.01) | |
| Psychological | Extent a patient enjoys life | 3.76(1.02) | 14.30(2.21) |
| | Extent a patient feels his/her life to be meaningful | 4.26(0.92) | |
| | Ability to concentrate | 3.89(1.03) | |
| | Ability to accept bodily appearance | 4.04(1.19) | |
| | Extent a patient has negative feelings such as blue mood, despair, anxiety, depression? (REVERSED) | 4.08(1.07) | |
| Level of Independence | Extent a patient needs any medical treatment to function in daily life (REVERSED) | 4.55(0.93) | 16.21(2.50) |
| | Ability to get around | 4.06(0.71) | |
| | Satisfaction with the ability to perform daily living activities | 3.81(0.93) | |
| | Satisfaction with capacity for work | 3.79(1.01) | |
| Social Relationships | Extent a patient feels accepted by the people he/she knows | 3.80(1.16) | 13.82(3.15) |
| | Satisfaction with personal relationships | 4.18(1.04) | |
| | Satisfaction with sex life | 1.96(1.26) | |
| | Satisfaction with the support a patient gets from friends | 3.88(1.22) | |
| Environmental | Extent a patient feels safe in daily life | 3.78(1.03) | 13.58(2.27) |
| | Extent a patient's physical environment is healthy | 3.69(1.02) | |
| | Having enough money to meet a patient day | 2.4(1.46) | |
| | Availability of information a patient needs in day-to-day life | 2.59(1.50) | |
| | Extent a patient has the opportunity for leisure activities | 3.36(1.03) | |
| | Satisfaction with the conditions of living place | 3.91(1.00) | |
| | Satisfaction with access to health services | 4.46(0.79) | |
| | Satisfaction with transport | 2.97(1.12) | |
| SRPB | Extent a patient is bothered by people blaming for his/her HIV status (REVERSED) | 4.38(1.18) | 17.13(3.57) |
| | Extent a patient fears the future (REVERSED) | 4.33(1.33) | |
| | Extent a patient worries about death (REVERSED) | 4.48(1.22) | |
| | Satisfaction with oneself | 3.96(0.98) | |
| Overall HRQOL | | 3.81(0.53) | 15.23(2.12) |

SRPB: Spirituality, Religion, Personal Beliefs

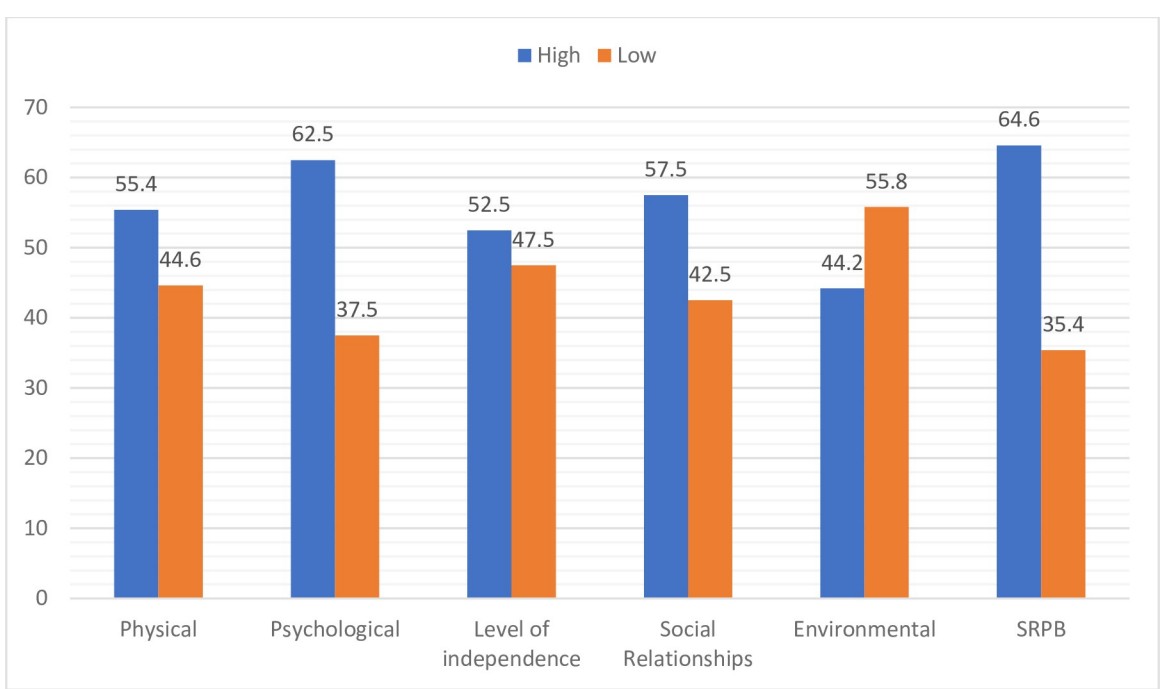

**Fig 1. Level of mean scores of QoL domains for PLWHA on HAART at MTUTH, March 2018.**

A multivariable logistic analysis was also fitted to determine predictors of QoL. Accordingly, patients with normal current health (AOR = 3.38, 95% CI = 1.56–7.31) and having family support (AOR = 3.12, 95% CI = 1.51–6.46) were found to have positive association with global score of QoL, while patients who having low HAART adherence (AOR = 0.40, 95%, CI = 0.19–0.86) had negative association with global score of QoL (Table 6).

## Discussion

The term QoL has been used to describe the overall sense of wellbeing with respect to happiness and general level of satisfaction with life. Given the longevity of life achievable with the current therapeutic strategies for PLWHA, QoL has emerged as a significant measure of health outcome, and quality of life enhancement as an important goal. The identification of factors that determine QoL is important to better tailor health and social care services, and thereby improve the functioning and wellbeing of people living with HIV.

Besides, determining the extent of QoL, this study found that factors like current health condition, family support and HAART adherence have been found to be strongly associated with the QoL of PLWHA. According to our study, more than half (57.1%%) of patients have a high global QoL. In contrary, studies from Bangladesh [35] reported global QoL score of low.

The study revealed that HAART adherence and QoL has a strong association. Patients who had a low/medium HAART adherence were 60% less likely to have a high global score of QoL compared with patients with high HAART adherence. A study done in South Africa [36] and Ethiopia [23] also reported the same findings that adherence is known to contribute to the QoL of PLWHA, as persons with greater ability to adhere to their ART regimens have better QoL. Adherence is found to improve clinical conditions of patients and suppress viral load [37–39] which in turn influence QoL.

According to the present study, current normal health had a positive association with the QoL. Hence, patients who reported normal current health were 3.38 times more likely to have

**Table 6. Multivariable logistic regression analysis of factors associated with QoL of PLWHA on HAART at MTUTH, March 2018.**

| Variables | Global Score of QoL | | Bivariate analysis | | Multivariable analysis | |
|---|---|---|---|---|---|---|
| | High (%) | Low (%) | P-value | COR (955 CI) | P-value | AOR (955 CI) |
| Sex | | | | | | |
| Male | 48 | 39 | - | 1.00 | | |
| Female | 89 | 64 | 0.65 | 0.88(0.52–1.50) | | |
| Age | | | | | | |
| ≤30 | 58 | 30 | 0.12 | 1.72(0.86–3.46) | 0.146 | 2.03(0.78–5.29) |
| 31–40 | 51 | 48 | 0.88 | 0.95(0.49–1.85) | 0.396 | 1.47(0.60–3.60) |
| >40 | 28 | 25 | - | 1.00 | | 1.00 |
| Education | | | | | | |
| Illiterate | 19 | 20 | 0.058 | 0.32(0.10–1.04) | 0.499 | 1.94(0.29–13.15) |
| Primary | 66 | 65 | 0.047 | 0.34(0.12–0.98) | 0.838 | 1.19(0.22–6.35) |
| Secondary | 37 | 13 | 0.931 | 0.95(0.29–3.13) | 0.344 | 2.16(0.44–10.69) |
| College | 15 | 5 | | 1.00 | | 1.00 |
| Occupation | | | | | | |
| Farmer* | 24 | 29 | | 1.00 | | 1.00 |
| Gov't employee | 31 | 11 | 0.006 | 3.40(1.42–8.17) | 0.979 | 0.98(0.23–4.12) |
| Private Business | 31 | 25 | 0.294 | 1.50(0.70–3.19) | 0.322 | 0.61(0.23–1.63) |
| Unemployed | 41 | 31 | 0.229 | 1.55(0.76–3.15) | 0.544 | 0.75(0.29–1.93) |
| Other | 10 | 6 | 0.232 | 2.01(0.64–6.34) | 0.949 | 0.95(0.20–4.43) |
| Marital status | | | | | | |
| Single | 12 | 6 | 0.100 | 2.77(0.82–9.31) | 0.299 | 2.44(0.45–13.10) |
| Married | 81 | 48 | 0.037 | 2.34(1.05–5.19) | 0.333 | 1.70(0.58–4.98) |
| Divorced | 31 | 31 | 0.463 | 1.38(0.58–3.30) | 0.960 | 0.97(0.31–3.04) |
| Widowed | 13 | 18 | | 1.00 | | 1.00 |
| Current ill health | | | | | | |
| Yes | 22 | 47 | | 1.00 | | 1.00 |
| No | 115 | 56 | 0.000 | 4.39(2.41–7.98) | 0.002 | 3.38(1.56–7.31) |
| Family support | | | | | | |
| Yes | 105 | 47 | 0.000 | 3.91(2.25–6.80) | 0.002 | 3.12(1.51–6.46) |
| No | 32 | 56 | | 1.00 | | 1.00 |
| Substance use (ever) | | | | | | |
| Yes | 38 | 36 | 0.23 | 0.71(0.41–1.24) | 0.53 | 0.78(0.36–1.70) |
| No | 99 | 67 | | 1.00 | | 1.00 |
| HAART adherence | | | | | | |
| High | 116 | 71 | | 1.00 | | 1.00 |
| Medium/low | 21 | 32 | 0.004 | 0.40(0.22–0.75) | 0.018 | 0.40(0.19–0.86) |
| OIs | | | | | | |
| Yes | 31 | 37 | 0.025 | 0.52(0.30–0.92) | 0.20 | 1.60(0.78–3.32) |
| No | 106 | 66 | | 1.00 | | 1.00 |
| Current BMI (Kg/m$^2$) | | | | | | |
| Under/Overweight | 48 | 52 | 0.013 | 0.51(0.30–0.87) | 0.15 | 0.61(0.32–1.19) |
| Normal BMI | 88 | 49 | | 1.00 | | 1.00 |

*Farmer and daily laborer

high global score of QoL as compared to patients who were currently ill. In line with this finding, the study conducted in Nigeria [40] revealed that participants who reported being

currently ill had poorer QoL in all the domains. The preoccupation of patients with diseases and the physical symptoms they experienced can impact negatively on QoL. In the absence of current illness, PLWHA can feel independent, self-reliance and physical and psychologically fit to execute their daily activities which helps them to feel positive about themselves.

From this finding, patients who get a continuous support from their family were 3.12 times more likely to have high global score of QoL. The positive impact of family support on patients QoL is also cited in previous studies [9, 41–43]. Family support is important for patients to feel secure, enhance self-confidence, minimize stress, and discrimination. The support from family largely depends on disclosure of sero-status, which is high in our study (86.25%). Disclosure is found to associated with QoL [44]. Family support can also help to improve adherence to HAART which in turn suppress viral load and improve clinical condition of patients, ultimately contribute for improved QoL.

The findings of study should be interpreted with consideration of the following limitations. The study is a single facility, cross-sectional design, hence extrapolation to other areas should be with precaution. The study may be underpowered to detect difference among groups especially variables with multiple categories. In addition, the study included a point in time data without follow-up. The other limitation is a measurement of adherence using MMAPS, which is subjective by its nature and has a chance of recall bias as participants were expected to remember two weeks lag time. In addition, since an interviewer administered questionnaire was used, there may be social desirability bias in sensitive information such as sex life, substance use, discrimination, and social relationships.

## Conclusions

Majority of the participants (57.1%) had a high global score of QoL. Normal current health and family support were associated with better global score of QoL, while low adherence was negatively associated with global score of QoL. Efforts should be strength to further improve quality of life PLWHA and further research should be done with longitudinal and qualitative designs to ascertain the findings.

## Supporting information

**S1 Data.**
(DOCX)

## Acknowledgments

We acknowledge PLWHA participated in this study for their cooperation during data acquisition. Our gratitude extended to health professionals and data clerk for their invaluable contributions during data collection.

## Author Contributions

**Conceptualization:** Addisu Desta, Tessema Tsehay Biru.

**Data curation:** Addisu Desta, Tessema Tsehay Biru, Adane Teshome Kefale.

**Formal analysis:** Addisu Desta, Tessema Tsehay Biru, Adane Teshome Kefale.

**Methodology:** Tessema Tsehay Biru.

**Supervision:** Addisu Desta.

**Writing – original draft:** Tessema Tsehay Biru, Adane Teshome Kefale.

**Writing – review & editing:** Addisu Desta, Tessema Tsehay Biru, Adane Teshome Kefale.

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
