## [Decision Letter · Decision Letter 0]

29 May 2020

PONE-D-20-05500

Health related quality of life of people receiving highly active antiretroviral therapy in Southwest Ethiopia

PLOS ONE

Dear Dr. Kefale,

Thank you for submitting your manuscript to PLOS ONE. After careful consideration, we feel that it has merit but does not fully meet PLOS ONE’s publication criteria as it currently stands. Therefore, we invite you to submit a revised version of the manuscript that addresses the points raised during the review process.

We look forward to receiving your revised manuscript.

Kind regards,

Haikel A. Lim, M.D., M.Sc.

Academic Editor

PLOS ONE

Journal Requirements:

2. We noticed you have some minor occurrence of overlapping text with the following previous publication, which needs to be addressed:

https://phcfm.org/index.php/phcfm/article/view/294/html

In your revision ensure you cite all your sources (including your own works), and quote or rephrase any duplicated text outside the methods section. Further consideration is dependent on these concerns being addressed."

3. Please address the following:

- Please describe how verbal consent was documented and witnessed.

- Please include additional information regarding the survey or questionnaire used in the study and ensure that you have provided sufficient details that others could replicate the analyses. For instance, if you developed a questionnaire as part of this study and it is not under a copyright more restrictive than CC-BY, please include a copy, in both the original language and English, as Supporting Information.

4. Thank you for including your ethics statement:  This study was conducted after formal letter was requested and obtained from the department of Pharmacy, Mizan Tepi University. Then the permission to collect data was obtained after official letters were submitted to the head of ART clinic.

Please amend your current ethics statement to confirm that your named institutional review board or ethics committee specifically approved this study.

Additional Editor Comments (if provided):

Thank you very much for your interest in submitting to PLOS One. The reviewers have raised valid points that I hope will be addressed in your revision of the manuscript.

In addition, please also address the following in your revision:

Please provide the participation rate for this study to better allow readers to appreciate the proportion of the sampling frame that agreed to review. This should also be included in the discussion as a limitation.

Please provide a reference for the QOL domain ranges of low/medium/high and the HAART score ranges of high/moderate/low adherence (page 6).

It is unclear if participants' pill counts were used or whether adherence was purely determined based on the MMAPS-8; please clarify.

Please spell out corrected and adjusted odds ratios before using the short forms (page 7).

Please highlight if all participants provided written informed consent, or if consent was waived (page 7).

The manuscript may benefit from another proof-read to correct the typographical and grammatical errors (e.g., the use of articles before nouns, etc.) throughout the manuscript.

Reviewers' comments:

Reviewer's Responses to Questions

**Comments to the Author**

1. Is the manuscript technically sound, and do the data support the conclusions?

Reviewer #1: Yes

Reviewer #2: Yes

2. Has the statistical analysis been performed appropriately and rigorously? 

Reviewer #1: Yes

Reviewer #2: Yes

3. Have the authors made all data underlying the findings in their manuscript fully available?

Reviewer #1: No

Reviewer #2: Yes

4. Is the manuscript presented in an intelligible fashion and written in standard English?

Reviewer #1: Yes

Reviewer #2: Yes

5. Review Comments to the Author

Reviewer #1: This is a manuscript that assess the QoL of PLWHA who were taking HAART at MTUTH and identify factors associated with QoL. However there are some major concerns that i will hope the authors can address:

1. How is the sampling done? is it convenience of random? the duration of mar 4 to apr 1 is of concern as well. it is a little too short combined with convenience sampling, the type of patients surveyed may have a strong responder bias.

2. Is the patient reported outcome like MMAPS and WHOQOL-BREF validated in your country?

3. There is much more females than males. is the proportion accurate of your country pls?

Thank you for the opportunity to review. Happy to review again.

Reviewer #2: I have the following comments and happy to review this paper again.

1) Under the Introduction, the authors needs to discuss recent global landmark studies on HIV and QoL . Please add the following at the end of second paragraph of the Introduction.

..... largely dependent on cultural, social and environmental contexts as well as individual perception

(12,13). QOL has been used as a criteria in assessing HIV/AIDS prevention programs, clinical treatment, and harm reduction strategies (Vu et al 2020). Multilevel interventions and long-term care, rehabilitation, behavioral therapy and social supports for patients receiving treatments, and application of e-health approaches can improve QOL of HIV patients (Tran et al 2020).

References:

Vu GT, Tran BX, Hoang CL, et al. Global Research on Quality of Life of Patients with HIV/AIDS: Is It Socio-Culturally Addressed? (GAPRESEARCH). Int J Environ Res Public Health. 2020;17(6):2127. Published 2020 Mar 23. doi:10.3390/ijerph17062127

Tran BX, Vu GT, Ha GH, et al. Global Mapping of Interventions to Improve the Quality of Life of People Living with HIV/AIDS: Implications for Priority Settings [published online ahead of print, 2020 Feb 12]. AIDS Rev. 2020;1‐15. doi:10.24875/AIDSRev.20000135

2) Under the Introduction, the authors stated "ART may indeed prolong life but may only do so at considerable cost to the QoL of PLWHA (15)". This statement requires further elaboration.

... at considerable cost to the QoL of PLWHA (15). There is a geographical discrepancy in HIV/AIDS economic evaluation research (Tran et al 2019). The high cost of ART might reduce the adherence to treatment and worsen QoL (Tran et al 2020). An improved QoL and resultant ability of the patient to resume normal life..

References

Tran BX, Nguyen LH, Turner HC, et al. Economic evaluation studies in the field of HIV/AIDS: bibliometric analysis on research development and scopes (GAPRESEARCH). BMC Health Serv Res. 2019;19(1):834. Published 2019 Nov 14. doi:10.1186/s12913-019-4613-0

Tran BX, Hoang CL, Tam W, et al. A global bibliometric analysis of antiretroviral treatment adherence: implications for interventions and research development (GAPRESEARCH). AIDS Care. 2020;32(5):637–644. doi:10.1080/09540121.2019.1679708.

3) Under the discussion, the authors stated "The current study found that patients who ever used substances like alcohol, cigarette and khat (AOR=0.39, 95% CI=0.19-0.82) has less likely to have high global quality of life score

compared to those patients who never used substances." Please add the following statement:

... , cigarette and khat (AOR=0.39, 95% CI=0.19-0.82). While drug-related risk behaviors were significantly reduced, alcohol and sex-related behaviors remained risk factors for HIV (Tran et al 2019). Similarly, previous studies had also reported such findings......

Reference:

Tran BX, Fleming M, Nguyen TMT, et al. Changes in Substance Abuse and HIV Risk Behaviors over 12-Month Methadone Maintenance Treatment among Vietnamese Patients in Mountainous Provinces. Int J Environ Res Public Health. 2019;16(13):2422. Published 2019 Jul 8. doi:10.3390/ijerph16132422.

6. PLOS authors have the option to publish the peer review history of their article (what does this mean?). If published, this will include your full peer review and any attached files.

Reviewer #1: No

Reviewer #2: Yes: Roger Ho

---

## [Author Response · Author response to Decision Letter 0]

8 Jun 2020

We thanks the reviewers and editor for their constructive comments helpful to improve the manuscript. Each concern is addressed and attached as a separate file.

---

## [Editor Report · Decision Letter 1]

10 Jun 2020

PONE-D-20-05500R1

Health related quality of life of people receiving highly active antiretroviral therapy in Southwest Ethiopia

PLOS ONE

Dear Dr. Kefale,

Thank you for submitting your manuscript to PLOS ONE. After careful consideration, we feel that it has merit but does not fully meet PLOS ONE’s publication criteria as it currently stands. Therefore, we invite you to submit a revised version of the manuscript that addresses the points raised during the review process.

I appreciate the time and effort that went into this revised manuscript. There are a few issues that still need to be addressed:

MAJOR

1. Please clarify this statement on page 6: "Patients scored 4-9.9, 10-14.9 and 15-20 were regarded as having low, medium and higher global scores of QoL[29, 30]". The cut-off scores are based on the Santos paper, not the Puri paper; and the Santos paper has used these scores based on the general (and not HIV-specific) WHOQOL on a sample of psychiatry graduates/residents. Please justify again the use of the cut-offs either by (a) suggesting that the Puri paper, and hopefully other papers, have adequately justified the use of these scores in spite of the contentious origin of the cutoffs, and discussing it in text either in the methods or discussion; (b) identifying other validated cut-off scores; or (c) using sample-specific cut-off scores of high/medium/low based on quartiles or tertiles or even a medium split and discussing this in text. Given this is the crux of the manuscript, this needs to be addressed before the manuscript can proceed.

MINOR

2. Please also clarify this statement on page 8: "We have approached 311 patients, but data of 240 was included in the analysis giving a response rate of 77.2%". (a) Please comment on how the study is adequately powered in spite of only 240 participants results were still found to be significant; and (b) Please comment on the reasons for non-participation/decline participation.

3. Please acknowledge in the discussion section the limitations of using the subjective self-reported MMAPS-8 vs. more objective measures like pill counts.

4. Please include in the methods that the data was collected via an interviewer-administered questionnaire and justify why it was not collected participant-completed self-reports; please also comment on this potential effect of social desirability bias (especially in the context of HIV/AIDS) on the discussion.

5. Please also comment on page 8 that the demographic distribution of participants approximate that of the sampling frame to allay any issues of biased sampling. 

We look forward to receiving your revised manuscript.

Kind regards,

Haikel A. Lim, MD, MSc

Academic Editor

PLOS ONE

Additional Editor Comments (if provided):

I appreciate the time and effort that went into this revised manuscript. There are a few issues that still need to be addressed:

MAJOR

1. Please clarify this statement on page 6: "Patients scored 4-9.9, 10-14.9 and 15-20 were regarded as having low, medium and higher global scores of QoL[29, 30]". The cut-off scores are based on the Santos paper, not the Puri paper; and the Santos paper has used these scores based on the general (and not HIV-specific) WHOQOL on a sample of psychiatry graduates/residents. Please justify again the use of the cut-offs either by (a) suggesting that the Puri paper, and hopefully other papers, have adequately justified the use of these scores in spite of the contentious origin of the cutoffs, and discussing it in text either in the methods or discussion; (b) identifying other validated cut-off scores; or (c) using sample-specific cut-off scores of high/medium/low based on quartiles or tertiles or even a medium split and discussing this in text. Given this is the crux of the manuscript, this needs to be addressed before the manuscript can proceed.

MINOR

2. Please also clarify this statement on page 8: "We have approached 311 patients, but data of 240 was included in the analysis giving a response rate of 77.2%". (a) Please comment on how the study is adequately powered in spite of only 240 participants results were still found to be significant; and (b) Please comment on the reasons for non-participation/decline participation.

3. Please acknowledge in the discussion section the limitations of using the subjective self-reported MMAPS-8 vs. more objective measures like pill counts.

4. Please include in the methods that the data was collected via an interviewer-administered questionnaire and justify why it was not collected participant-completed self-reports; please also comment on this potential effect of social desirability bias (especially in the context of HIV/AIDS) on the discussion.

5. Please also comment on page 8 that the demographic distribution of participants approximate that of the sampling frame to allay any issues of biased sampling.

---

## [Author Response · Author response to Decision Letter 1]

16 Jul 2020

The response is attached as a separate file

---

## [Editor Report · Decision Letter 2]

20 Jul 2020

Health related quality of life of people receiving highly active antiretroviral therapy in Southwest Ethiopia

PONE-D-20-05500R2

Dear Dr. Kefale,

We’re pleased to inform you that your manuscript has been judged scientifically suitable for publication and will be formally accepted for publication once it meets all outstanding technical requirements.

Kind regards,

Haikel A. Lim, MD, MSc

Guest Editor

PLOS ONE

Additional Editor Comments (optional):

Thank you for your revised manuscript. I am pleased to convey that this manuscript is ready for publication in PLOS ONE. Thank you once again for your submission and professionalism throughout the review process. I wish you the best of luck in your future research endeavours.
---

## [Editor Report · Acceptance letter]

22 Jul 2020

PONE-D-20-05500R2 

Health related quality of life of people receiving highly active antiretroviral therapy in Southwest Ethiopia 

Dear Dr. Kefale:

I'm pleased to inform you that your manuscript has been deemed suitable for publication in PLOS ONE. Congratulations! Your manuscript is now with our production department. 

Kind regards, 

on behalf of

Dr. Haikel A. Lim 

Guest Editor

PLOS ONE